# Prevalence of family-based elder abuse and its associated factors in Gandaki Province of Western Nepal: A cross-sectional study

Bharat Kafle[1*], Aman Shrestha[2,3], Saruna Ghimire[4], Preeti Bhattarai[1], Pratik Bhattarai[5], Amod Kumar Poudyal[6]

**1** School of Public Health, Karnali Academy of Health Sciences, Jumla, Karnali Province, Nepal, **2** Division of Gerontology, Department of Epidemiology & Public Health, School of Medicine, University of Maryland Baltimore, Baltimore, Maryland, United States of America, **3** Department of Sociology, Anthropology, and Public Health, University of Maryland Baltimore County, Baltimore, Maryland, United States of America, **4** Department of Sociology and Gerontology and Scripps Gerontology Center, Miami University, Oxford, Ohio, United States of America, **5** Department of Medicine, Manipal College of Medical Sciences, Pokhara, Nepal, **6** Central Department of Public Health, Institute of Medicine, Tribhuvan University, Maharajgunj, Kathmandu, Bagmati Province, Nepal

☯ These authors contributed equally to this work.
* bharatkafle29@gmail.com

## Abstract

### Background

Elder abuse is a rarely discussed public health and human rights issue in Nepal, largely due to traditional values emphasizing reverence for parents. These cultural norms can discourage recognition and reporting, especially within families. Consequently, nationwide or large-scale data on this issue is lacking. This study aims to estimate the prevalence and explore the factors associated with family-based elder abuse in Gandaki province, western Nepal.

### Methods

A cross-sectional design was conducted interviewing 612 participants (≥ 60 years) in household settings. The study areas, representing mountain, hill and tarai regions of Gandaki province, were randomly selected using multistage stratified sampling process. The dependent variable was measured using 17-item elder abuse scale. Multivariable logistic regression explored the factors associated with elder abuse.

### Results

More than half of the participants were from Nawalpur (57.2%), urban residents (68.1%), female (57.2%), without education (79.8%), and lived in multigenerational households (73.0%). The overall prevalence of elder abuse was reported at 56.4%, with caregiver neglect (50.8%) being the most prevalent sub-type. The prevalence was higher among females (66.6%) than males (42.8%). In adjusted multivariable

**Data availability statement:** All relevant files are available from the OpenICPSR repository and can be download using the link: https://doi.org/10.3886/E209442V1.

**Funding:** The author(s) received no specific funding for this work.

**Competing interests:** The authors have declared that no competing interests exist.

logistic regression, those who were female (AOR = 2.56, 95%CI: 1.64–4.01), older than 70 years (AOR = 1.53, 95%CI: 1.03–2.29), reluctant to disclose health issues to family members (AOR = 2.13, 95%CI: 1.36–3.34), believed in traditional healers (AOR = 1.89, 95%CI: 1.28–2.77) and lived in nuclear households (AOR = 1.85, 95%CI: 1.19–2.89) had higher odds of elder abuse. Those living in rural areas (AOR = 0.39, 95%CI: 0.25–0.61), having formal education (AOR = 0.31, 95%CI: 0.10–0.95), and having good self-reported health (AOR = 0.23, 95%CI: 0.12–0.46) were associated with lower odds of elder abuse.

## Conclusions

The study highlights a high prevalence of elder abuse, particularly caregiver neglect, with a disproportionate impact on women. Policy recommendations include raising awareness, strengthening legal protections, and improving caregiver training to meet these challenges effectively.

---

## Introduction

Elder abuse is a pervasive, deeply ingrained global social issue with serious consequences for individuals and society. While terms like "mistreatment" and "abuse" are often used interchangeably, this study adopts "abuse" for consistency. In general, elder abuse refers to any intentional or negligent act that puts an older adult at risk, perpetrated by individuals who have control over, are responsible for, or have a trusted, caring relationship with the older adult [1]. Defined by the World Health Organization as harmful act within a trusted relationship [2], elder abuse can manifest in various forms, including physical, mental, sexual, or emotional, and extend to financial exploitation, acts of neglect, and abandonment, such as denying involvement in decision-making processes and restricting personal spaces [1]. Elder abuse occurs both in institutional and community settings, leading to severe health impacts and reduced quality of life [3]. Beyond individuals, elder abuse also imposes considerable financial and caregiving burdens on society and the country [4].

Globally, a systematic review and meta-analysis, including studies from 28 countries, revealed that one in six older adults reported experiencing abuse in community settings [5]. In institutionalized settings, 64.2% of staff have admitted to abusing older adults [6]. However, these studies also indicate that the actual prevalence of elder abuse may be much higher due to factors like underreporting and the lack of a standardized definition and measurement methods for elder abuse. In South Asia, cross-sectional studies indicate elder abuse prevalences at 14% in Bangladesh [7], between 5% to 50% in India [8] and 54% in Pakistan [9]. Abuse towards older adults can result in severe physical and mental health issues, financial problems, and social consequences. Some of these include physical injuries, cognitive decline, depression, hospitalization, premature mortality, financial devastation, and forced placement in alternative care options [10].

Elder abuse, particularly by family members, is a significant yet overlooked issue in countries with traditional culture and strong filial piety (traditional values that emphasize reverence for parents). Nepal, a South Asian country nestled between India and China, is no exception [11]. On the one hand, elder abuse is perceived as a societal taboo in Nepal; on the other hand, the traditional belief systems that emphasize respect and care for older family members paradoxically create barriers to acknowledging, reporting, and addressing incidents of elder abuse [11,12]. While often perceived as rare in Nepali society, small-scale studies in the last ten years report a high prevalence between 46.6% to 65.6%, with caregiver neglect, up to 57.5%, being the most common subtype [11–14]. Additional forms of elder abuse reported in Nepal include accusations of witchcraft, caste-based discrimination, familial victimization, misuse of old-age allowances, negligence, sexual misconduct, and torture, often stemming from perceptions of older members as financial burdens on their families [15].

Nepal lacks a specific elder abuse policy but has a few legislations, regulations, and guidelines geared toward protecting its older population. The Senior Citizen Act (2006) ensures social security, welfare, and safeguarding rights for older adults and penalizes abuses [16]. The article 41 of the Constitution of Nepal (2015) ensures dignity, safety and social security [17]. The Domestic Violence Act (2009) criminalizes all forms of abuse within a household [18]. Additional efforts include the National Plan of Action for Senior Citizens (2005) [19], the Social Security Act (2018) [20], and the sixteenth five-year national plan [21], none of which have, in and of themselves, been adequate to address the problems of elder abuse in Nepal.

Although previous studies have shed light on the issue of elder abuse, they were limited by small sample size and geographical coverage. So far, no comprehensive national, provincial, or pooled prevalence study on elder abuse has been conducted in Nepal. The recent census data showing a significant increase in the older population of 60 years and above — from 6.4% in 2001 to 10.2% in 2021 — indicates Nepal's population is aging [22]. This demographic shift implies that issues concerning older adults, including incidents of elder abuse, are likely to become more prominent, underscoring the need for greater attention to this often-underreported issue. To the authors' knowledge, this study is the first to comprehensively examine elder abuse using representative samples from the mountain, hill and tarai regions within Gandaki Province, Nepal. The study aims to estimate the prevalence of elder abuse by family members and explore the factors associated with such abuse.

## Materials and methods

### Study design and settings

The study employed a community-based cross-sectional design, utilizing a quantitative approach for data collection and analysis. Gandaki Province (S1 Fig) was purposively selected for this study due to its high proportion of older adults, the highest among all seven provinces [23,24], its ethnic diversity with significant populations of "Brahaman/Chhetri" and "Adivasi/Janajati," as well as a notable presence of "Dalit/Muslim" minority groups [25]. The province spans all three ecological regions of Nepal — mountains, hills, plains, or tarai — offering diverse geographical and lifestyle contexts.

S1 Table presents various demographic and socioeconomic indicators for Gandaki Province and its study districts, comparing them to the national average using data from the 2021 census. Briefly, Gandaki Province has a total population of 2,466,427, accounting for 8.5% of the national population [22]. Approximately 13.5% of the population in this province are 60 years or older. The old-age dependency ratio in the province is 21.25, which is higher than the national old-age dependency ratio of 16.48 [22]. Regarding development, the Human Development Index in Gandaki Province ranges from 0.48 to 0.58, closely matching the national average of 0.49 [25].

### Study population

The study population included older adults who were at least 60 years at the time of the survey. Participants were required to be permanent resident of the study area and to have proficiency in the Nepali language. Those with hearing or speech impairments that interfered with effective communication were excluded from this study.

## Sample size and sampling

The sample size was calculated using one proportion formula, $\frac{(Z_{1-\alpha}+Z_{1-\beta})^2\ p(1-p)}{\varepsilon^2}$ inbuilt in Stata/MP 13.1 [26]. The reported prevalence of elder abuse was determined at 49.1% (i.e., p = 0.491) based on a previous study [11]. The power of the study was set at 80% (i.e., $Z_{1-\beta}$ is 0.84) with a 95% confidence interval (i.e., $Z_{1-\alpha}$ is 1.96) and 10% desired margin of error (i.e., $\varepsilon$ is 0.10). This initially resulted in a sample size of 194. However, after accounting for design effect of 1.5 and stratification factor of two, the sample size was recalculated to be 582. Finally, considering a 5% non-response rate, the final sample size was determined to be 613, which was rounded to 616 after sampling allocation, as illustrated in S2 Table.

Multistage stratified cluster sampling was used to select participants. In the first sampling stage, one district was randomly selected from each ecological region: Manang from the mountains, Tanahu from the hills, and Nawalpur, also called East Nawalparasi, from the tarai. Subsequently, using simple random sampling, one urban and one rural municipality were selected from the Nawalpur and Tanahu districts. However, in Manang, only one rural municipality was chosen randomly, as this district has no urban municipality. In the subsequent stage, within each selected municipality, wards (the smallest administrative units in Nepal) were randomly chosen as clusters or primary sampling units based on population proportionate to size. Then, within each ward, systematic random sampling was used to select every third household until the target sample size of 22 households per cluster was reached. The sampling flow diagram of the study is presented below (Fig 1). Only one older adult from each selected household was interviewed. For more than one eligible older adult living in the same household, only one was selected at random.

## Survey instruments

A semi-structured questionnaire was first developed in English and then translated into Nepali to facilitate data collection. To ensure accuracy and linguistic consistency, bilingual experts conducted a back translation from Nepali to English. Three co-authors in this manuscript and two local research assistants from Gandaki Province, all with prior experience in research data collection, were recruited for the fieldwork. They underwent a one-day training session that covered various aspects of the study, including its purpose, research protocols and procedures, ethical considerations, participant recruitment, interviewing techniques and safety measures. A pilot testing of the survey instrument was carried out with 30 older adults in ward no. 26 of Kathmandu Metropolitan City. While the responses from this pilot test were excluded from the primary data analysis, they proved valuable for identifying and rectifying several sensitive language issues in the questionnaire.

## Data collection

A paper-based questionnaire was used to interview 612 older adults through face-to-face, in-person interviews conducted between January 9, 2020, and February 26, 2020. Given the sensitive nature of the study, each interview was conducted in private space within the participant's home to ensure confidentiality. To minimize interruptions from family members, a second research assistant was present during the interview to provide any necessary support. The collected data were subsequently digitized using data entry panels created in EpiData 3.1 [27].

## Study variables

**Dependent variable.** Older adults participating in the study were asked to report any instances of abuse by family members in the past three months using a set of 21 questions, each measured with a binary "Yes" or "No" response. The abuse scale, developed for older adults in eastern Nepal, has been previously used to assess abuse among older adults in community settings within Nepal [11,14]. Operational definitions of abuse and its various types are detailed in the S3 Table. The questionnaire covered various types of abuse, starting with an assessment of elder abuse in general, followed by three items on physical abuse, five items on psychological abuse and caregiver neglect, three items on financial abuse,

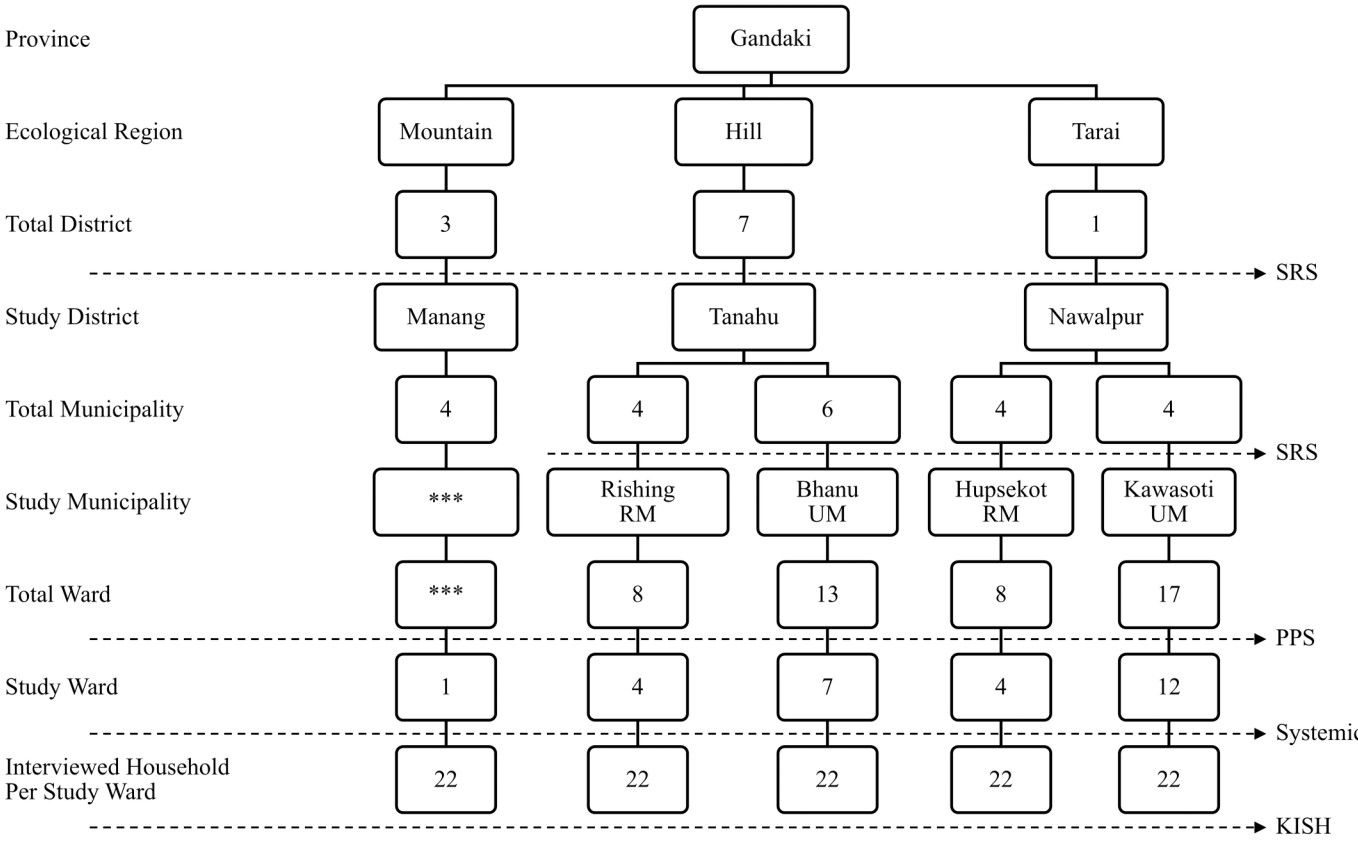

***Specific information are not disclosed due to small population size to protect individual privacy and comply with ethical guidelines.

**Fig 1. Sampling procedure of the study.** *Note.* RM *Note.* RM=rural municipality; UM=urban municipality; SRS=simple random sampling; PPS=probability proportional to size; KISH=a random sampling technique.

one item on sexual abuse and three items of legal abuse. Comprehensive details of the tool and its 21core items can be found in previous studies [11,14]. Some example questions include: "Have you been beaten by any members of your family?", "Does your family insult you in front of others?", and "Do you feel that your family members are uncommunicative or unresponsive towards you?" As this study focused on elder abuse within family settings, the three questions related to legal abuse were excluded, as they were contextually outside family settings. Furthermore, the initial question, "Have you experienced some form of mistreatment in the last three months?" was omitted, considering that similar information could be inferred from other items in the scale. In this study, elder abuse was defined as the occurrence of at least one of the five types of abuse captured in the tool: physical, psychological, financial, sexual abuse, and caregiver neglect. The internal consistency of the scale, which included the seventeen items used in this study, resulted in a high McDonald's Omega score of 0.91, signifying the high reliability of this tool [28].

**Independent variables**

**Sociodemographic factors.** Sociodemographic factors included a range of variables, including study districts, municipality type, age, gender, ethnicity, marital status, and disability status. As described in the methods section above, the three randomly selected districts were Manang, Tanahu, and Nawalpur, representing the mountain, hill, and tarai

regions in order. The classification of selected municipalities as "rural" or "urban" was based on the Nepal Gazette 2073, an official publication of the Government of Nepal [29]. Age was a numeric variable that displayed a right-skewed distribution and thus was divided into "60 to 70 years" and "above 70 years." Gender had two categories: "female" and "male." Following the approach of previous studies [11,14], ethnicity was reclassified into three categories: "Brahaman/Chhetri," "Adivasi/Janajati," and "Dalit/Muslim" [30]. The "Brahaman/Chhetri" included "Hill Brahaman/Chhetri" and "Tarai/Madhesi Brahaman/Chhetri." The "Adivasi/Janajati" groups included "Newar," "Hill Janajati," and "Tarai Janajati." Similarly, "Dalit/Muslim" included "Hill Dalit," "Tarai/Madhesi Dalit," and "Musalman."

Marital status was recategorized into two groups: "married" and "without a partner"; the latter category included individuals who were "unmarried," "widowed," "divorced," or "separated." Self-reported disability status was defined following the definition provided in the schedule related to sub-section (1) of section 3 in Nepal Gazette 2074 [31]. The definition follows: "A person with a disability means a person who has long-term physical, mental, intellectual or sensory disability or functional impairments or existing barriers that may hinder his or her full and effective participation in social life on an equal basis with others." Participants were asked if they had "no disability," "physical disability," "visual impairments," "hearing impairments," "hearing and visual impairments," "unable to speak," "psychosocial disabilities," "intellectual disability," "hemophilia," "autism," or "multi-disability." Participants with at least one disability were grouped as "with disability." Those without disabilities were grouped as "without disability."

**Socioeconomic factors.** Socioeconomic factors included participants' education status and two family-related factors: (1) type of family ("nuclear" vs. "multigenerational"); and (2) monthly family income (converted to US$; the average exchange rate in 2020 was $1 = NRs. 118.5106). Education was grouped into three categories: "no education," "non-formal education," and "formal education." "No education" referred to participants who could not read and write; those who went to classes or training outside of formal school settings and thus could read and write in Nepali were categorized as "non-formal education," while those who went to structured and systematic learning in schools were categorized as "formal education."

**Social participation and interaction.** Participation in social activities was assessed by asking participants if they got involved in any community-organized social activities and events within one year. Similarly, participation in community programs was assessed by asking whether they participated or gathered in any program organized by the community within the last week. Visits from relatives in the past month were measured by asking whether any relative had visited the participant within the last month. Participants could respond "yes" or "no" to each question.

**Health-related factors.** Healthcare utilization was assessed by asking participants if they had visited a medical institution in the past year, with binary responses of "yes" or "no." Behavioral health factors included two items: 1) whether participants believed in traditional healers and 2) their reluctance to disclose health issues to family members, both with binary responses of "yes" or "no." Self-reported health status was measured by asking participants, "How do you rate your health condition today?" The possible responses were: "good," "satisfactory," "neutral," "not satisfactory," and "poor." Due to low-frequency counts, related categories were merged to create a three-level self-rated health variable: "poor" (combining "poor" and "not satisfactory"), "good" (combining "good" and "satisfactory"), and "neutral." Healthcare access was assessed by asking if participants had access to the health facility to receive healthcare services. The responses were "yes" or "no."

## Data analyses

Data from EpiData was imported into SAS 9.4 for analysis [32]. In descriptive analyses (Table 1), categorical variables were presented as frequencies and percentages. The only continuous independent variable, monthly family income, was reported as the median and interquartile range due to its high right skew. Table 2 shows the prevalence of different types of elder abuse and their gender-specific prevalence and differences assessed using the Pearson Chi-square test of independence. Similarly, the bivariate relationships between abuse experience and categorical independent variables were

**Table 1. Overall characteristics of study participants (N = 612).**

| Variables | Frequency | Percentage |
|---|---|---|
| **Sociodemographic factors** | | |
| Study district | | |
| Nawalpur | 350 | 57.2 |
| Tanahu | 240 | 39.2 |
| Manang | 22 | 3.6 |
| Municipality type | | |
| Urban | 417 | 68.1 |
| Rural | 195 | 31.9 |
| Age | | |
| 60-70 years | 347 | 56.7 |
| Above 70 years | 265 | 43.3 |
| Gender | | |
| Male | 262 | 42.8 |
| Female | 350 | 57.2 |
| Ethnicity | | |
| Brahaman/Chhetri | 252 | 41.2 |
| Adivasi/Janajati | 284 | 46.4 |
| Dalit/Muslim | 76 | 12.4 |
| Marital status | | |
| Married | 385 | 62.9 |
| [1]Without a partner | 227 | 37.1 |
| Disability status | | |
| Without disability | 344 | 56.2 |
| With disability | 268 | 43.8 |
| **Socioeconomic factors** | | |
| Education | | |
| No education | 488 | 79.8 |
| Non-formal education | 98 | 16.0 |
| Formal education | 26 | 4.2 |
| Type of family | | |
| Multigenerational | 447 | 73.0 |
| Nuclear | 165 | 27.0 |
| [2]Monthly family income in US$ (Median±IQR) | 126.6 | 185.6 |
| **Social participation and interaction** | | |
| [2]Participation in social activities | | |
| No | 356 | 58.2 |
| Yes | 256 | 41.8 |
| [3]Participation in community programs | | |
| No | 364 | 59.5 |
| Yes | 248 | 40.5 |
| Visits from relatives in the past month | | |
| Yes | 484 | 79.1 |
| No | 128 | 20.9 |
| **Health related factors** | | |
| Healthcare visit within the last year | | |
| Yes | 448 | 73.2 |

*(Continued)*

**Table 1.** (Continued)

| Variables | Frequency | Percentage |
|---|---|---|
| No | 164 | 26.8 |
| Believe traditional healer | | |
| Yes | 311 | 50.8 |
| No | 301 | 49.2 |
| Reluctance to disclose health issues to family members | | |
| No | 441 | 72.1 |
| Yes | 171 | 27.9 |
| Self-Rated health | | |
| Poor | 94 | 15.3 |
| Neutral | 334 | 54.6 |
| Good | 184 | 30.1 |
| [2]Healthcare access | | |
| Yes | 489 | 79.9 |
| No | 123 | 20.1 |

*Note:* Md = median, IQR = inter-quartile range; [1]includes unmarried, widowed, divorced and separated; [2] excluded in regression based on AIC criteria; [3] excluded in regression due to multicollinearity.

**Table 2.** Overall and gender-specific prevalence of various forms of elder abuse (N = 612).

| Type of abuse | Prevalence (%) (95%CI) | | | p-value |
|---|---|---|---|---|
| | Total | Male | Female | |
| Physical | 22.7 (19.45-26.24) | 10.7 (7.22-15.07) | 31.7 (26.87-36.87) | **<0.001***** |
| Psychological | 35.5 (31.66-39.39) | 21.4 (16.57-26.84) | 46.0 (40.69-51.38) | **<0.001***** |
| Caregiver neglect | 50.8 (46.78-54.85) | 35.9 (30.07-42.01) | 62.0 (56.91-67.09) | **<0.001***** |
| Financial | 11.9 (9.47-14.76) | 11.5 (7.86-15.94) | 12.3 (9.04-16.19) | 0.752 |
| Sexual | 4.3 (2.79-6.16) | 2.3 (0.84-4.92) | 5.7 (3.53-8.69) | **0.038*** |
| **Overall** | **56.4 (52.34-60.34)** | **42.8 (36.68-48.98)** | **66.6 (61.36-71.50)** | **<0.001***** |

*Note:* *$p < 0.05$, **$p < 0.01$, ***$p < 0.001$, indicating statistical significance based on the Pearson Chi-square Test of Association between abuse and gender.

assessed using the Pearson Chi-square test of independence, while the Mann-Whitney test evaluated the relationship between elder abuse and the continuous independent variable, monthly family income (Table 3).

All the variables described in Table 3 were assessed for multicollinearity using the variance influence factor (VIF) with a cutoff point 2.5 in the initial model [33]. Notably, "Participation in social activities" (VIF = 2.82) and "participation in community programs" (VIF = 2.96) showed evidence of multicollinearity, with a high correlation (r = 0.79). As a result, the variable with the higher VIF value (i.e., participation in community programs) was removed from the model to address multicollinearity concerns.

The best-fitting model, determined by minimizing the Akaike Information Criterion (AIC), excluded six variables: districts, ethnicity, marital status, monthly family income, participation in social activities, and healthcare access. Considering

**Table 3. Bivariate analyses of characteristics of study participants (N = 612).**

| Variables | Experienced abuse? | | p-value |
|---|---|---|---|
| | **Yes (n,%)** | **No (n,%)** | |
| | **34 (56.4%)** | **267 (43.6%)** | |
| **Sociodemographic factors** | | | |
| Study District | | | 0.776 |
| Nawalpur | 196 56.0) | 154 (44.0) | |
| Tanahu | 138 (57.5) | 102 (42.5) | |
| Manang | 11 (50.0) | 11 (50.0) | |
| Municipality type | | | **<0.001*** |
| Urban | 257 (61.6) | 160 (38.4) | |
| Rural | 88 (45.1) | 107 (54.9) | |
| Age | | | 0.114 |
| 60-70 years | 186 (53.6) | 161 (46.4) | |
| Above 70 years | 159 (60.0) | 106 (40.0) | |
| Gender | | | **<0.001*** |
| Male | 112 (42.8) | 150 (57.2) | |
| Female | 233 (66.6) | 117 (33.4) | |
| Ethnicity | | | 0.674 |
| Brahaman/Chhetri | 143 (56.8) | 109 (43.2) | |
| Adivasi/Janajati | 156 (54.9) | 128 (45.1) | |
| Dalit/Muslim | 46 (60.5) | 30 (39.5) | |
| Marital status | | | **<0.001*** |
| Married | 195 (50.6) | 190 (49.4) | |
| [1]Without a partner | 150 (66.1) | 77 (33.9) | |
| Disability status | | | **0.001**** |
| Without disability | 173 (50.3) | 171 (49.7) | |
| With disability | 172 (64.2) | 96 (35.8) | |
| **Socioeconomic factors** | | | |
| Education | | | **<0.001*** |
| No education | 296 (60.7) | 192 (39.3) | |
| Non-formal education | 44 (44.9) | 54 (55.1) | |
| Formal education | 5 (19.2) | 21 (80.8) | |
| Type of family | | | 0.142 |
| Multigenerational | 244 (54.6) | 203 (45.4) | |
| Nuclear | 101 (61.2) | 64 (38.8) | |
| [2]Monthly family income in US$ (Median±IQR) | 109.7 (194.1) | 168.8 (210.9) | **<0.001***u** |
| **Social participation and interaction** | | | |
| [2]Participation in social activities | | | **<0.001*** |
| No | 226 (63.5) | 130 (36.5) | |
| Yes | 119 (46.5) | 137 (53.5) | |
| [3]Participation in community programs | | | **<0.001*** |
| No | 229 (62.9) | 135 (37.1) | |
| Yes | 116 (46.8) | 132 (53.2) | |
| Visits from relatives in the past month | | | **0.006**** |
| Yes | 259 (53.5) | 225 (46.5) | |
| No | 86 (67.2) | 42 (32.8) | |

*(Continued)*

**Table 3.** (Continued)

| Variables | Experienced abuse? | | p-value |
|---|---|---|---|
| | Yes (n,%) | No (n,%) | |
| | 34 (56.4%) | 267 (43.6%) | |
| **Health related factors** | | | |
| Healthcare visit within the last year | | | <0.001*** |
| Yes | 276 (61.6) | 172 (38.4) | |
| No | 69 (42.1) | 95 (57.9) | |
| Believe traditional healer | | | <0.001*** |
| Yes | 196 (63.0) | 115 (37.0) | |
| No | 149 (49.5) | 152 (50.5) | |
| Reluctance to disclose health issues to family members | | | <0.001*** |
| No | 215 (48.8) | 226 (51.2) | |
| Yes | 130 (76.0) | 41 (24.0) | |
| Self-Rated health | | | <0.001*** |
| Poor | 77 (81.9) | 17 (18.1) | |
| Neutral | 187 (56.0) | 147 (44.0) | |
| Good | 81 (44.0) | 103 (56.0) | |
| ²Healthcare access | | | 0.497 |
| Yes | 279 (57.1) | 210 (42.9) | |
| No | 66 (53.7) | 57 (46.3) | |

*Note:* *p<0.05, **p<0.01, ***p<0.001; ¹includes unmarried, widowed, divorced and separated; ²excluded in regression based on AIC criteria; ³excluded in regression due to multicollinearity; IQR: inter-quartile range; ᵘP-value from Mann-Whitney test, all others are from Pearson Chi-square test.

the distinctions between the three studied districts and the importance of ethnicity and marital status in defining and describing elder abuse in Nepali society [11,34,35], district, ethnicity, and marital status were deemed crucial and were retained in the final model. Therefore, the final adjusted model (Table 4) included district, municipality type, age, gender, ethnicity, marital status, disability status, education, family type, relative visiting within the last month, healthcare visit within the last year, belief in traditional healers, reluctance to disclose health issues to family members and self-rated health. All of these variables were treated as confounders and controlled in the regression analyses to account for their possible influence on the relationships between independent and dependent variables.

Concordance statistics (c-statistic), which range from 0.5 to 1 (with higher values indicating a better fit), were used to evaluate the model's goodness of fit. The c-statistic for the final adjusted model was 0.78, signifying that the final model demonstrated good predictive power [36]. Furthermore, advanced diagnostics did not detect any influential observations [37]. Both unadjusted and adjusted multivariable logistic regression models were conducted to explore the factors associated with elder abuse. Odds ratios (OR) and 95% confidence intervals (CI) from both unadjusted and adjusted multivariable logistic regression models are presented in Table 4. These values indicate the odds of experiencing any form of elder abuse at different levels of the independent variables, both with and without controlling for covariates. Statistical significance was defined as a p-value less than 0.05.

## Ethical consideration

This study obtained ethical approval from the Institutional Review Committee at the Institute of Medicine, Tribhuvan University (reference no. 194/(6–11) E2 076/077) and received written permission from the Social Development Ministry of Gandaki Province (reference no. 1019/075–076) in Nepal. Participants were informed about the study

**Table 4. Unadjusted and adjusted odds ratio (OR) for factors associated with elder abuse using multivariable logistic regression.**

| Variables | Unadjusted OR (95%CI) | ¹Adjusted OR (95%CI) |
|---|---|---|
| **Sociodemographic factors** | | |
| **Study district (ref = "Nawalpur")** | | |
| Tanahu | 1.06 (0.76-1.48) | 0.93 (0.61-1.43) |
| Manang | 0.79 (0.33-1.86) | 1.54 (0.55-4.28) |
| **Municipality type (ref = "Urban")** | | |
| Rural | **0.51 (0.36-0.72)***** | **0.39 (0.25-0.61)***** |
| **Age** (ref = "60 to 70 years") | | |
| Above 70 years | 1.30 (0.94-1.80) | **1.53 (1.03-2.29)*** |
| **Gender** (ref = "Male") | | |
| Female | **2.67 (1.92-3.71)***** | **2.56 (1.64-4.01)***** |
| **Ethnicity** (ref = "Brahaman/Chhetri") | | |
| Adivasi/Janajati | 0.93 (0.66-1.31) | 1.02 (0.67-1.55) |
| Dalit/Muslim | 1.17 (0.69-1.97) | 0.96 (0.52-1.78) |
| **Marital status** (ref = "Married") | | |
| ²Without a partner | **1.90 (1.35-2.67)***** | 1.19 (0.78-1.82) |
| **Disability status** (ref = "Without disability") | | |
| With disability | **1.77 (1.28-2.46)***** | **1.63 (1.10-2.42)*** |
| **Socioeconomic factors** | | |
| **Education** (ref = "No education") | | |
| Non-formal education | **0.53 (0.34-0.82)**** | 1.09 (0.62-1.90) |
| Formal education | **0.15 (0.06-0.42)***** | **0.31 (0.10-0.95)*** |
| **Type of family** (ref = "Multigenerational") | | |
| Nuclear | 1.31 (0.91-1.89) | **1.85 (1.19-2.89)**** |
| **Social participation and interaction** | | |
| **Visits from relatives in the past month** (ref = "Yes") | | |
| No | **1.78 (1.18-2.68)**** | **1.64 (1.01-2.70)*** |
| **Health related factors** | | |
| **Healthcare visit within the last year** (ref = "Yes") | | |
| No | **0.45 (0.32-0.65)***** | 0.66 (0.43-1.01) ‡ |
| **Believe traditional healer** (ref = "No") | | |
| Yes | **1.74 (1.26-2.40)***** | **1.89 (1.28-2.77)**** |
| **Reluctance to disclose health issues to family members** (ref = "No") | | |
| Yes | **3.33 (2.24-4.96)***** | **2.13 (1.36-3.34)**** |
| **Self-Rated health** (ref = "Poor") | | |
| Neutral | **0.28 (0.16-0.50)***** | **0.31 (0.17-0.59)***** |
| Good | **0.17 (0.09-0.32)***** | **0.23 (0.12-0.46)***** |

*Note.* *p < 0.05, **p < 0.01, ***p < 0.001, ‡approached significance; ¹The model is mutually adjusted for all the variables presented in Table 3; ²includes unmarried, widowed, divorced and separated.

and its purpose, voluntary participation, and their rights to withdraw anytime. They provided written consent before their interviews, either through signatures or thumbprints. For participants unable to read, they discussed the consent forms with their trusted individuals before providing their consent. Interviews were conducted privately in the participants' homes, with arrangements made to ensure confidentiality and minimize interruptions. No proxy interviews were conducted.

## Results

### Characteristics of study participants

Table 1 below shows the overall characteristics of the study participants. More than half of the older adults were from the Nawalpur (57.2%), residents of urban municipalities (68.1%), between 60 and 70 years of age (56.7%), and were female (57.2%). Participants from "Brahaman/Chhetri" groups (41.2%) and "Adivasi/Janajati" (46.4%) comprised the majority, with only 12.4% belonging to "Dalit/Muslim." About 63% were married, 44% reported having one or more disabilities, 80% did not have any education, and 73% lived in multigenerational households. The median monthly family income was $126.6 (IQR = 185.6). More than half did not engage in social activities (58.2%) and community programs (59.5%), and about 21% had no recent visits from relatives. Approximately three-fourths (73.2%) sought healthcare, although more than half (50.8%) also believed in traditional healers. Notably, 27.9% were reluctant to disclose health issues to family, but only 15.3% rated their health as poor; 54.6% reported neutral health.

### Prevalence of elder abuse and its sub-types

Table 2 below presents data on the prevalence of family-based elder abuse in Gandaki Province, Nepal. The overall prevalence of elder abuse was 56.4% (95%CI: 52.34–60.34). Caregiver neglect was the most reported form of elder abuse (50.8%), followed by psychological abuse (35.5%), while sexual abuse (4.3%) was the least reported type. The prevalence of overall abuse and all its subtypes, except for financial abuse, was statistically significant and higher among females than males (Table 2).

### Bivariate association between participants' characteristics and their experience of abuse

Table 3 below depicts the characteristics of the study participants and bivariate comparisons based on abuse experience. Significant independent variables under bivariate tests were municipality type, gender, marital status, disability status, education, monthly family income, participation in social activities and community programs, visits from relatives in the past month, healthcare visit within the last year, believe traditional healer, reluctance to disclose health issues to family members and self-rated health.

The prevalence of elder abuse was significantly higher among those who lived in urban municipalities (61.6%), were females (66.6%), without a partner (66.1%), and with disability (64.2%). With education, elder abuse was found to significantly drop from no education at 60.7% to formal education at 19.2%. Abuse was higher among participants who did not participate in social activities (63.5%) or community programs (62.9%) and those who did not have visits from relatives in the past month (67.2%). Participants who were reluctant to disclose health issues to family members (76.0%) and had poor self-rated health (81.9%) faced significantly higher abuse.

### Factors associated with elder abuse: Regression analyses

The results of the multivariable logistic regression analysis for factors associated with family-based elder abuse are presented in Table 4. After controlling for all other variables in the model, residing in a rural municipality was associated with 61% lower odds of elder abuse than living in an urban municipality (AOR = 0.39, 95%CI: 0.25–0.61). Similarly, older females had 2.56 times the odds of experiencing abuse than older males (95%CI: 1.64–4.01). Those with disabilities reported 63% higher odds of abuse than those without disabilities (AOR = 1.63, 95%CI: 1.10–2.42). In the unadjusted model, education showed significance for both formal and non-formal categories. However, only the formal education category retained significance when controlled for covariates and was associated with 69% lower odds of experiencing abuse (AOR = 0.31, 95%CI: 0.10–0.95). Older adults living in nuclear families reported 85% higher odds of abuse compared to those living in multigenerational families (AOR = 1.85, 95%CI: 1.19–2.89).

Among health-related factors, believing in a traditional healer, reluctance to disclose health issues to family members, and self-rated health showed highly significant associations with elder abuse after controlling the covariates. Older adults who believed in a traditional healer reported 1.89 times the odds of abuse compared to those who did not believe (95%CI: 1.28–2.77). Participants who were reluctant to disclose their health issues to family members had more than twice the increased odds of abuse compared to those who openly discussed their health issues (AOR: 2.13, 95%CI: 1.36–3.34). Regarding self-rated health, older adults who rated their health as neutral had 69% lower odds of experiencing elder abuse (AOR = 0.31, 95%CI: 0.17–0.59), while those who rated their health as good had 77% lower odds (AOR = 0.23, 95%CI: 0.12–0.46), in comparison to older adults who rated their health as poor.

## Discussion

This study examined the prevalence of elder abuse in Gandaki Province and explored the associated factors. Over half of the participants reported experiencing abuse, with a significantly higher prevalence among females. Furthermore, participants who were over 70 years old, reluctant to disclose health issues to family members, believed in traditional healers, or lived in nuclear households had higher odds of reporting elder abuse compared to their respective counterparts. Conversely, those living in rural areas, with formal education, or reporting good self-rated health had lower odds of experiencing elder abuse.

The high prevalence of abuse reported by participants is consistent with previous cross-sectional studies in Nepal, which reported prevalences ranging from 46.6% to 65.6% [11–14,38]. In contrast, a 2017 systematic review and meta-analysis on elder abuse in community settings estimated the global prevalence of elder abuse at 15.7% [5]. This indicates that the burden of elder abuse found in this study is approximately 3.6 times higher than the global average. Additionally, recent literature from neighboring countries has reported varying elder abuse prevalence, ranging from 5% to 50% in India and 0.2% to 64% in China [8,39]. Differences in prevalence among studies may arise from cross-national socio-economic and cultural variations, but they also highlight potential discrepancies in methodology and the tools used to measure elder abuse. Notably, the assessment tools used in studies conducted in Nepal lacked validation within the local context. Nepal's aging population faces additional challenges, including weak enforcement of protection laws for older adults and limited awareness of elder abuse both within families and across society, which further exacerbates the issue.

Despite historically being a covert issue, this study's findings, along with earlier research from Nepal, highlight the severity and pervasiveness of elder abuse in the country. The elevated prevalence of elder abuse in Nepal may also stem from cultural values such as filial piety and the typical multigenerational family structures, where older family members often rely on their adult children for care without receiving compensation [14]. This can result in financial strain and heightened caregiving demands on families. Furthermore, the government of Nepal offers limited support for its older population, with existing welfare schemes providing minimal assistance. Also, institutional care for older adults is scarce in the country due to the cultural principle of filial piety, which mandates that adult children take care of their older parents legally and morally. The few existing facilities tend to be expensive, making them unaffordable for most older adults. Moreover, there is a strong stigma associated with seeking institutional care, as it is perceived as a sign of being abandoned by one's family, further discouraging older adults from accessing such services. Consequently, older family members primarily rely on their families for care without compensation. Daughters-in-law typically take on the primary role of providing daily tasks and personal assistance, while sons focus on ensuring financial stability [40]. This traditional caregiving structure burdens family members significantly, potentially resulting in financial strain and intensified caregiving demands. In line with the theorized strain plausibility, neglect by family caregivers was the most common form of elder abuse in this study, accounting for 50.8% of cases, consistent with previous findings from Nepal [11–14]. The increased dependency and declining financial autonomy in old age align with the political-economic theory of elder abuse [41], which states that older adults become vulnerable to potential abuse from family members as their role in the family shifts following retirement from the workforce.

Abuse was reported more frequently by older individuals who were above 70 years compared to those between 60 and 70 years, a trend supported by prior studies from Nepal [42] and a scoping review of international research indicating an increase in elder abuse with age [43]. This finding may be explained by a decline in general health and well-being, as well as functional limitations that come with aging, leading to increased dependence on caregivers for daily activities and subsequently placing an increased burden on them, which may contribute to instances of abuse. Consistent with this explanation, this study also found that both disability and poor self-reported health—measures of physical functioning and overall health—were associated with higher odds of experiencing abuse. The findings align with a previous study from Nepal that found older adults with disabilities were approximately 12 times more vulnerable to abuse [42]. Similarly, older individuals who rated their health better had lower odds of elder abuse, echoing findings from an Indian longitudinal study that linked self-reported poor health with a 64% higher risk of elder abuse [44]. Taken together, the burden of caregiving, particularly when required without sufficient support and financial compensation, as in Nepal, can lead to a rise in instances of abuse.

In this study, older females had 2.56 times the odds of experiencing elder abuse than older males; two earlier studies from Nepal support these findings [12,13]. However, the literature on gender differences in elder abuse presents mixed results in both Nepal and global settings. Some research from Nepal reports higher abuse among older males [3], while two other studies found no statistically significant gender differences in abuse experiences [11,14]. Consistent with our findings, a 2017 meta-analysis found a higher prevalence of elder abuse among females in Asia and the Pacific, although the association was not statistically significant [5]. Although most studies suggest a trend of increased abuse among older women, the lack of statistical significance may be due to small sample sizes, as the meta-analysis included only 52 studies, and the two prior non-significant results from Nepal were derived from small-scale studies with only 212 and 339 participants. The observed gender disparity in abuse experiences may be attributed to patriarchal norms in Nepali families and is supported by feminist theories of elder abuse. The feminist theory of elder abuse explains this phenomenon through the perspective of traditional power dynamics within families and societies [41]. Within this framework, it is observed that older women often wield less influence and authority, contributing to their increased vulnerability to abuse. The age gap between married couples is common in Nepal, where women tend to be younger than their male partners. This trend is shaped by long-standing cultural traditions, societal perceptions, and individual choices. In most cases, women assume caregiving roles for their husbands in their later years, but upon their husbands' passing, they find themselves reliant on their adult children for support. Consequently, Nepali women may encounter increased vulnerability, particularly considering their longer life expectancy than their male counterparts. Furthermore, it is unfortunate that Nepali society has historically undervalued women's contributions, perpetuating their perception as a dependent population throughout their life journey. This high dependence may contribute to the elevated prevalence of abuse among older women in Nepal.

Two related findings—higher odds of abuse in urban areas and within nuclear families—may be correlated due to the tendency for households in urban areas to adopt more nuclear family structures. Although there is a lack of comparative studies on urban-rural differences in elder abuse in Nepal, the findings from this study represent a pioneering step in assessing these variations in reported abuse experiences. Research conducted in Canada and Nigeria corroborates the findings of this study [43]. However, the findings from India, which shares a similar socio-cultural context with Nepal, consistently reported higher elder abuse in rural settings [44,45] and thus do not support the conclusion of this study. Nevertheless, a study from India reported physical abuse higher among rural residents, while neglect and psychological abuse were higher among urban residents [46]. With urban migration emerging as a recent phenomenon in Nepal, older adults often relocate to where their children secure employment, exposing them to potential abuse in unfamiliar surroundings. Despite previous studies in India indicating lower instances of elder abuse in urban settings attributed to higher levels of education among older populations [47], only 4% of participants in this study had received formal education. Further, although there is a potential interaction between family type and urban-rural settings that could influence abuse experiences, the moderation analysis in this study did not yield significant results. Thus, this area warrants further exploration

in future research, particularly in light of the recent reclassification of urban areas in Nepal. Recently, the urban area coverage defined in 2017 was 66.08%, which was only 17% in 2011, and this reclassification fails to adhere to internationally accepted standards of defining urban areas [48], suggesting a potential misclassification between urban and rural municipalities.

Abuse experiences reported in this study are more common in nuclear family settings, aligning with earlier findings in Nepal that smaller family sizes are associated with a higher report of abuse [42]. A cross-country study from Bangladesh with similar family structures supports these conclusions [7], suggesting that strong family solidarity, filial piety, and collectivism may contribute to lower elder abuse in multigenerational households. Additionally, in multigenerational family settings, financial and caregiving responsibilities for older family members are shared among multiple family members, which helps to alleviate financial stress and caregiver burnout, subsequently reducing the risk of abuse. However, larger household sizes may also result in more family conflicts, potentially increasing the risk of elder abuse. Therefore, this complex area warrants further investigation in future research.

Although distinct, the findings on the significant association of education and belief in traditional healers could be explained with a common underlying role of awareness and decision-making. Regarding education, this study highlights the significant role of education in lowering the elder abuse experience. Those with formal education experienced 69% lower odds of facing elder abuse. Prior studies in Nepal also support the strong linkage between education and lower elder abuse [13,14]. For instance, a study by Yadav and colleagues found 69% higher odds of elder abuse among older individuals who had no education compared to those with education [14]. Education thus can serve as a vital intervention, enabling individuals to recognize mistreatment and access legal assistance when necessary. Higher levels of education offer individuals better access to resources, information, support systems, and coping mechanisms, which can help reduce overall risk and vulnerability to abuse. Education also empowers individuals, contributing to a lower likelihood of experiencing abuse. This study uncovered an interesting link between elder abuse and belief in traditional healers, with older adults who trusted these healers experiencing higher rates of abuse. The relationship between elder abuse and faith in traditional healing practices has not been extensively studied, making direct comparisons with existing research challenging. However, this trend may be associated with the fact that older adults who have limited education often depend on traditional healers, reflecting the same underlying factors that link abuse to a lack of education [49].

Reluctance to share health concerns with family members, which was significantly linked to elder abuse, poses a notable health risk for older adults in Nepal and can lead to serious health problems. In the absence of prior studies on this subject, cross-comparison is limited. This hesitation may stem from older adults' desire to avoid burdening their children or reflect deeper trust issues and internal family conflicts. When older parents do not disclose their health problems to family members, it can result in misunderstandings, unrealistic expectations, and a lack of empathy, potentially creating conditions where neglect and abuse may thrive.

## Limitations and strengths

The cross-sectional nature of the data in this study limits its ability to make definitive conclusions about causality. Since elder abuse was evaluated based on self-reported instances from the past three months, there is a possibility of recall and information bias. Although the reliability score was high in this study, the tool used to measure elder abuse was not validated, which could affect the study's overall rigor. Additionally, the study did not measure the frequency or severity of the abuse experienced by older individuals within their families. Future research should prioritize using validated tools to assess elder abuse, as well as redefine elder abuse criteria to include measures of its frequency or severity.

This study represents a pioneering effort, covering all three ecological regions of Gandaki province, Nepal and employing a rigorous sampling approach. To the author's knowledge, this is the first study in Nepal to examine the disparities in elder abuse between urban and rural environments.

## Conclusions

This study provides valuable insight into the higher prevalence of elder abuse in Gandaki province, with caregiver neglect appearing as the most common abuse type. Despite being a sensitive societal issue, the findings emphasize the importance of openly discussing and raising awareness about elder abuse in family settings. Abuse can severely impact the health and well-being of older adults, highlighting the need for caregiver education to increase awareness and understanding of the issue. Additionally, interventions should target modifiable risk factors such as caregiver burden, anxiety, and lack of social support while providing respite care and counseling services to prevent abuse and improve older adults' outcomes. Healthcare professionals should regularly assess signs of abuse, caregiver burden, anxiety, and strained family relationships when interacting with older adults and their caregivers.

Public awareness campaigns are necessary to inform the public about the prevalence, risk factors, and consequences of elder abuse by family members. Promoting positive attitudes towards aging and challenging ageist stereotypes can also help prevent abuse. The study stresses the urgency of establishing and enforcing a comprehensive legal framework to combat issues of elder abuse effectively. While creating policies to protect older adults from abuse is essential, their successful implementation is even more crucial, though this is often lacking in countries like Nepal due to political instability and economic challenges. However, there is hope as the United Nations outlines five priorities to combat elder abuse within the decade of healthy aging from 2021 to 2030, and this study offers baseline information to guide federal and provincial governments and stakeholders in that effort.

## Supporting information

**S1 Tables. Demographic and socioeconomic characteristics of study districts.**
(DOCX)

**S1 Fig. Geographical map of Nepal with study districts in Gandaki Province.**
(TIF)

## Acknowledgments

We acknowledge the support from the Social Development Ministry and the local-level authorities in Gandaki province throughout this study. We are also deeply thankful to all the participants who generously contributed their time and insight to this research endeavor.

## Author contributions

**Conceptualization:** Bharat Kafle, Aman Shrestha, Saruna Ghimire, Preeti Bhattarai.

**Data curation:** Bharat Kafle, Aman Shrestha, Preeti Bhattarai.

**Formal analysis:** Bharat Kafle, Aman Shrestha, Saruna Ghimire.

**Investigation:** Bharat Kafle, Preeti Bhattarai, Pratik Bhattarai.

**Methodology:** Bharat Kafle, Aman Shrestha.

**Project administration:** Bharat Kafle, Preeti Bhattarai.

**Resources:** Bharat Kafle.

**Software:** Aman Shrestha.

**Supervision:** Amod Kumar Poudyal.

**Validation:** Bharat Kafle, Aman Shrestha.

**Visualization:** Aman Shrestha.

**Writing – original draft:** Bharat Kafle, Aman Shrestha, Saruna Ghimire.

**Writing – review & editing:** Bharat Kafle, Aman Shrestha, Saruna Ghimire, Preeti Bhattarai, Pratik Bhattarai, Amod Kumar Poudyal.

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
