## [Decision Letter · Decision Letter 0]

4 Sep 2024

PONE-D-24-19113Prevalence of family-based elder abuse and its associated factor in Gandaki province of western Nepal: a cross-sectional studyPLOS ONE

Dear Dr. Kafle,

Thank you for submitting your manuscript to PLOS ONE. After careful consideration, we feel that it has merit but does not fully meet PLOS ONE’s publication criteria as it currently stands. Therefore, we invite you to submit a revised version of the manuscript that addresses the points raised during the review process.

Please go through both reviewers' comments and address their queries.  Please clearly write methodology as suggested by reviewer 2 .

We look forward to receiving your revised manuscript.

Kind regards,

Shalik Dhital, PhD

Academic Editor

PLOS ONE

Journal requirements: 1. When submitting your revision, we need you to address these additional requirements. Please ensure that your manuscript meets PLOS ONE's style requirements, including those for file naming. The PLOS ONE style templates can be found at https://journals.plos.org/plosone/s/file?id=wjVg/PLOSOne_formatting_sample_main_body.pdf and https://journals.plos.org/plosone/s/file?id=ba62/PLOSOne_formatting_sample_title_authors_affiliations.pdf. 2. We note that you have indicated that there are restrictions to data sharing for this study. PLOS only allows data to be available upon request if there are legal or ethical restrictions on sharing data publicly. For more information on unacceptable data access restrictions, please see http://journals.plos.org/plosone/s/data-availability#loc-unacceptable-data-access-restrictions.  Before we proceed with your manuscript, please address the following prompts: a) If there are ethical or legal restrictions on sharing a de-identified data set, please explain them in detail (e.g., data contain potentially identifying or sensitive patient information, data are owned by a third-party organization, etc.) and who has imposed them (e.g., a Research Ethics Committee or Institutional Review Board, etc.). Please also provide contact information for a data access committee, ethics committee, or other institutional body to which data requests may be sent. b) If there are no restrictions, please upload the minimal anonymized data set necessary to replicate your study findings to a stable, public repository and provide us with the relevant URLs, DOIs, or accession numbers. For a list of recommended repositories, please seehttps://journals.plos.org/plosone/s/recommended-repositories. You also have the option of uploading the data as Supporting Information files, but we would recommend depositing data directly to a data repository if possible. We will update your Data Availability statement on your behalf to reflect the information you provide. 3. We note that your Data Availability Statement is currently as follows: [All relevant data are within the manuscript and its Supporting Information files.] Please confirm at this time whether or not your submission contains all raw data required to replicate the results of your study. Authors must share the “minimal data set” for their submission. PLOS defines the minimal data set to consist of the data required to replicate all study findings reported in the article, as well as related metadata and methods (https://journals.plos.org/plosone/s/data-availability#loc-minimal-data-set-definition). For example, authors should submit the following data: - The values behind the means, standard deviations and other measures reported;- The values used to build graphs;- The points extracted from images for analysis. Authors do not need to submit their entire data set if only a portion of the data was used in the reported study. If your submission does not contain these data, please either upload them as Supporting Information files or deposit them to a stable, public repository and provide us with the relevant URLs, DOIs, or accession numbers. For a list of recommended repositories, please see https://journals.plos.org/plosone/s/recommended-repositories. If there are ethical or legal restrictions on sharing a de-identified data set, please explain them in detail (e.g., data contain potentially sensitive information, data are owned by a third-party organization, etc.) and who has imposed them (e.g., an ethics committee). Please also provide contact information for a data access committee, ethics committee, or other institutional body to which data requests may be sent. If data are owned by a third party, please indicate how others may request data access. 4. We notice that your supplementary tables are included in the manuscript file. Please remove them and upload them with the file type 'Supporting Information'. Please ensure that each Supporting Information file has a legend listed in the manuscript after the references list.

Reviewers' comments:

Reviewer's Responses to Questions

**Comments to the Author**

1. Is the manuscript technically sound, and do the data support the conclusions?

Reviewer #1: Yes

Reviewer #2: No

2. Has the statistical analysis been performed appropriately and rigorously? 

Reviewer #1: Yes

Reviewer #2: No

3. Have the authors made all data underlying the findings in their manuscript fully available?

Reviewer #1: Yes

Reviewer #2: Yes

4. Is the manuscript presented in an intelligible fashion and written in standard English?

Reviewer #1: Yes

Reviewer #2: No

5. Review Comments to the Author

Reviewer #1: The authors have well-articulated the manuscript and addressed an important topic. This study brings attention to elder mistreatment, which is often concealed in the study area due to the prevailing culture of filial piety.

Introduction

This section is well-written with adequate literature review.

Methods section

The authors have well-described methods using appropriate techniques and procedures. I have some minor comments, which are as follows:

• Since this study was conducted during peak time of COVID-19 pandemic situation, I suggest the authors to briefly explain how the team handled this situation and managed to face-to-face interview 612 interviews.

• Can you please mention one or two examples in the set of 21 questions of abuse? How were those questioned asked to the participants?

• In the Data analyses section, the authors mentioned SAS version 9.04 while it should have been 9.4. I suggest the authors verify the version of SAS.

• The method section could end up with mentioning how you interpret the results, particularly for regression model, e.g. OR, and at level of ?? significance or mention the CI%.

Results:

Well-done with your results! I have minor comment, to start your results with participants’ profile first and then interpret the prevalence of elder abuse, followed by regression analyses.

Discussion

A brief summary of your results including the associated factors would be informative to the readers.

Approximately 56% of the participants reported experiencing abuse. It is possible that this value is overestimated due to the pandemic situation, particularly regarding caregiver neglect (50.8%). The COVID-19 pandemic likely exacerbated these factors due to fears of infection, social distancing measures, job losses, and reduced services for older adults. I suggest authors to discuss these viewpoints as well.

Reviewer #2: This is interesting research who are familiar with the topic. This reviewer has following observations for further revision.

1. Structured abstract is needed.

2. In the introduction section, review of context specific literature- total elder population, policy and program context in Nepal, issues and challenges of elder population, rationale and significance of study is not coming through. Needs more lit review and synthesis with relevant research gaps for this research.

3. It is unclear how sample was estimated and how they were selected in the data collection process. A flow chart can be useful to show them.

4. It is unclear how tool was designed, tested and ensure validity of the tool in Nepali context.

5. Major issue lies in the analysis output, for example, the analysis of prevalence of categories of variables should be row percentage than column percent while running analysis of experience of abuse. Total prevalence of abuse in Hill region is estimated 138*100/240=57.5%. if yes is included there is no need to write no % and numbers.

6. After careful analysis, result section needs revision.

7. Once we have clear findings, discussion section needs focused analysis and interpretation linking with policy, practices, systems and services in the federalized context of Nepal.

6. PLOS authors have the option to publish the peer review history of their article (what does this mean? ). If published, this will include your full peer review and any attached files.

**Do you want your identity to be public for this peer review?** For information about this choice, including consent withdrawal, please see our Privacy Policy .

Reviewer #1: No

Reviewer #2: No

---

## [Author Response · Author response to Decision Letter 0]

8 Oct 2024

Responses attached in the 'Response to Reviewers' document.

---

## [Editor Report · Decision Letter 1]

19 Nov 2024

PONE-D-24-19113R1Prevalence of family-based elder abuse and its associated factor in Gandaki province of western Nepal: A cross-sectional studyPLOS ONE

Dear Dr. Kafle,

Thank you for submitting your manuscript to PLOS ONE. After careful consideration, we feel that it has merit but does not fully meet PLOS ONE’s publication criteria as it currently stands. Therefore, we invite you to submit a revised version of the manuscript that addresses the points raised during the review process.

Please refer to the Additional Editor Comments below. 

We look forward to receiving your revised manuscript.

Kind regards,

Emma Campbell, Ph.D

Staff Editor

PLOS ONE

on behalf of:

Shalik Ram Dhital, PhD

Academic Editor

PLOS ONE

Additional Editor Comments:

Dear Kafle,

I hope you are doing well. I am writing regarding your manuscript titled. After careful review, we have identified several major revisions that need to be addressed. I have raised the following significant points that require attention. Please see each comment in Track Change version of your manuscript. Please revise the manuscript to address these comments and provide a detailed response letter explaining how you have handled each point. This will help us understand the changes you've made and expedite the review process. In addition to addressing these major revisions, we ask that you ensure your manuscript fully complies with PLOS ONE's author guidelines. These guidelines cover aspects such as manuscript formatting, ethical considerations, and data availability. Once you have made the necessary revisions, please submit the updated manuscript through the PLOS ONE submission system. Kindly include a summary of the changes made, along with a point-by-point response to the reviewers' comments.

Thank you for your efforts and your contribution to the field. We look forward to receiving your revised manuscript and moving forward with the review process.

With kind regards

Shalik Ram Dhital, PhD

Academic Editor

PLOS ONE

---

## [Author Response · Author response to Decision Letter 1]

31 Jan 2025

Response are available in the 'Response to Reviewers' document.

---

## [Editor Report · Decision Letter 2]

13 Apr 2025

Prevalence of family-based elder abuse and its associated factors in Gandaki province of western Nepal: A cross-sectional study

PONE-D-24-19113R2

Dear Dr. Kafle,

We’re pleased to inform you that your manuscript has been judged scientifically suitable for publication and will be formally accepted for publication once it meets all outstanding technical requirements.

Kind regards,

Jianhong Zhou

Staff Editor

PLOS ONE
---

## [Editor Report · Acceptance letter]

PONE-D-24-19113R2

PLOS ONE

Dear Dr. Kafle,

I'm pleased to inform you that your manuscript has been deemed suitable for publication in PLOS ONE. Congratulations! Your manuscript is now being handed over to our production team.

Kind regards,

on behalf of

Dr. Jianhong Zhou

Staff Editor

PLOS ONE